# Small nucleoli are a cellular hallmark of longevity

Varnesh Tiku[1,2], Chirag Jain[1], Yotam Raz[3], Shuhei Nakamura[4], Bree Heestand[5], Wei Liu[6], Martin Späth[2], H. Eka. D. Suchiman[3], Roman-Ulrich Müller[2,7], P. Eline Slagboom[3], Linda Partridge[1,2] & Adam Antebi[1,2]

Animal lifespan is regulated by conserved metabolic signalling pathways and specific transcription factors, but whether these pathways affect common downstream mechanisms remains largely elusive. Here we show that NCL-1/TRIM2/Brat tumour suppressor extends lifespan and limits nucleolar size in the major *C. elegans* longevity pathways, as part of a convergent mechanism focused on the nucleolus. Long-lived animals representing distinct longevity pathways exhibit small nucleoli, and decreased expression of rRNA, ribosomal proteins, and the nucleolar protein fibrillarin, dependent on NCL-1. Knockdown of fibrillarin also reduces nucleolar size and extends lifespan. Among wildtype *C. elegans*, individual nucleolar size varies, but is highly predictive for longevity. Long-lived dietary restricted fruit flies and insulin-like-peptide mutants exhibit small nucleoli and fibrillarin expression, as do long-lived dietary restricted and IRS1 knockout mice. Furthermore, human muscle biopsies from individuals who underwent modest dietary restriction coupled with exercise also display small nucleoli. We suggest that small nucleoli are a cellular hallmark of longevity and metabolic health conserved across taxa.

[1] Max Planck Institute for Biology of Ageing, Joseph Stelzmann Strasse 9b, 50931 Cologne, Germany. [2] Cologne Excellence Cluster on Cellular Stress Responses in Aging-Associated Diseases (CECAD), University of Cologne, 50674 Cologne, Germany. [3] Section of Molecular Epidemiology, Leiden University Medical Center, 2300 RC Leiden, The Netherlands. [4] Department of Genetics, Graduate School of Medicine, Osaka University 2-2 Yamadaoka, Suita 565-0871, Japan. [5] Lineberger Comprehensive Cancer Center, University of North Carolina, Chapel Hill, North Carolina 27599-3280, USA. [6] Department of Molecular and Cellular Biology, Huffington Center on Aging, Baylor College of Medicine, Houston, Texas 77030, USA. [7] Department II of Internal Medicine and Center for Molecular Medicine Cologne, University of Cologne, 50674 Cologne, Germany. Correspondence and requests for materials should be addressed to A.A. (email: antebi@age.mpg.de).

Over the last several decades, studies in model genetic organisms have revealed that animal lifespan is plastic and regulated by evolutionarily conserved signalling pathways. These pathways include reduced insulin/IGF and mTOR signalling, reduced mitochondrial function, dietary restriction mediated longevity, and signals from the reproductive system, which act through specific constellations of transcription factors to extend life[1]. Whether they converge on common regulators or shared downstream processes, however, has remained largely an open question. One process universally required across the major longevity pathways is autophagy, the turnover of cellular components through lysosomal degradation[2,3]. Accordingly, a key transcriptional regulator of autophagy, HLH-30/TFEB has been shown to be responsible to extend life in various *C. elegans* longevity pathways[4]. More recently, we and others have shown the Mondo complexes to also do so, as part of an extensive HLH transcriptional network together with HLH-30/TFEB[5,6]. However, the full extent of this regulatory tier and the precise relationship to downstream processes remain poorly understood.

A related question is whether there are common causal biomarkers of aging. Considerable efforts have been invested to identify biomarkers predictive of biological age, including physiologic readouts, metabolic parameters, glycomic profiles and others[7]. Nevertheless, markers with strong predictive power, and those proximal to the process of aging have remained elusive. More recently, the discovery of a DNA methylation clock, which monitors changes in hundreds of sites across the genome, has been used to robustly predict human chronological age, as well as aspects of biological age, but the functional and physiologic significance of this marker still remains obscure[8].

The nucleolus is a nuclear subcompartment where ribosomal RNA is synthesized and assembled into ribosomal subunits[9]. It is a dynamic organelle subject to inputs from growth signalling pathways, nutrients, and stress, whose size correlates with rRNA synthesis[10]. The nucleolus is also a production site for other ribonucleoprotein particles, including various splicing factors, the signal recognition particle, stress granules and the siRNA machinery. It thus can be thought of as a central hub of protein and RNA quality control and assembly.

Here we report the discovery of the nucleolus as a convergent point of regulation of major longevity pathways across species. Our studies reveal that several *C. elegans* longevity pathways impinge on regulators of nucleolar function, including NCL-1, a homologue of BRAT/TRIM2, which inhibits production of FIB-1/fibrillarin, a nucleolar protein involved in the regulation and maturation of rRNA. Our work suggests that small nucleoli are a visible cellular hallmark of longevity and metabolic health, and that molecules associated with nucleolar function might serve as predictive, causal biomarkers of life expectancy.

## Results

***ncl-1* mediates DR and other forms of longevity**. We identified the conserved B-box protein NCL-1 in genetic screens for novel mediators of DR induced longevity[11]. NCL-1 is an ortholog of the TRIM2/BRAT tumour suppressor, which inhibits rRNA transcription and protein synthesis[12]. Consistent with a role in ribosome biogenesis, NCL-1 regulates nucleolar size and *ncl-1* mutants have larger nucleoli especially in neuronal, muscle and hypodermal cells[13]. We found that whereas *ncl-1* loss had little effect on wildtype lifespan, it potently suppressed the longevity of *eat-2* mutants, a genetic model of DR (Fig. 1a and Supplementary Fig. 1a). *ncl-1* mutation also abrogated longevity across a wide range of bacterial food dilutions, revealing a function in the nutrient response to dietary restriction (Fig. 1b and Supplementary Fig. 1b).

We next asked if *ncl-1* also modulates longevity in other known longevity models. Reduced TOR signalling is partly responsible for lifespan extension under DR conditions[14]. Accordingly, *ncl-1* mutation abrogated longevity induced by *let-363*/TOR RNAi knockdown (Fig. 1c and Supplementary Fig. 1c), suggesting that *ncl-1* mediates lifespan extension on TOR down-regulation. Reduced insulin/IGF signalling potently promotes longevity across taxa, and knockdown of *daf-2*, the *C. elegans* insulin/IGF receptor, doubles the lifespan[15]; *ncl-1* mutation partially suppressed *daf-2* longevity as well (Fig. 1d and Supplementary Fig. 1d). Furthermore, *ncl-1* loss abolished lifespan extension in long-lived germlineless *glp-1* mutants[16] (Fig. 1e) and partially suppressed longevity triggered by mutation of the iron sulfur protein *isp-1*, which reduces mitochondrial function[17] (Fig. 1f). A modest reduction in translation is known to extend lifespan in different organisms[18]. *C. elegans* harbouring loss-of-function mutations in *ife-2* or *ifg-1*, which encode translation initiation factors, have reduced translation and extended lifespan[14,19,20]. Similarly *rsks-1* codes for the ribosomal protein S6 kinase (S6K), which is a known downstream target of the TOR kinase whose deficiency reduces protein synthesis and extends lifespan in multiple species[21]. Loss of *ncl-1* by RNAi largely abolished the longevity phenotype of *ife-2, ifg-1* and *rsks-1* mutant worms (Fig. 1g and Supplementary Fig. 1e,f). Altogether these findings reveal that *ncl-1* works in major longevity pathways to affect lifespan, as part of a convergent mechanism.

To investigate the role of *ncl-1* further, we generated extra-chromosomal transgenic lines expressing wildtype *ncl-1* fused to *gfp*. Arrays restored normal nucleolar size and extended lifespan in *eat-2;ncl-1* double mutants, demonstrating that the transgene is functional (Supplementary Fig. 1g–j). The fusion protein was found to reside in multiple tissues including neurons, body wall muscle, pharynx, seam cells and vulva (Supplementary Fig. 1k). Consistent with an instructive role, *ncl-1* over-expression in the wildtype background was sufficient to reduce nucleolar size (Supplementary Fig. 1g,h) and increase lifespan in two independent transgenic lines (Fig. 1h). No further increase of lifespan of *eat-2* on *ncl-1* over-expression was seen, indicating an overlapping mechanism (Supplementary Fig. 1l).

**Nucleolar size inversely correlates with longevity**. Since *ncl-1* affects both longevity and nucleolar size, we wondered if nucleolar size also changes in long-lived genotypes. To address this issue, we measured the nucleolar size of superficial hypodermal cells on the first day of adulthood. As previously shown, *ncl-1* mutants had enlarged nucleoli compared to wildtype (Fig. 2a and Supplementary Fig. 2a). We further observed that *eat-2* mutants had smaller nucleoli (Fig. 2a,c), and accordingly found that reducing nutrient levels through bacterial dilution diminished nucleolar size (Fig. 2b). Nucleolar size was enlarged in *eat-2;ncl-1* double mutants, revealing that *ncl-1* is epistatic to *eat-2* for both nucleolar size and longevity (Fig. 2a and Supplementary Fig. 2b). These intriguing observations led us to ask whether other longevity pathways more generally affect nucleolar size. Surprisingly, reduced insulin/IGF signalling (*daf-2*), reduced mTOR (*let-363*), reduced mitochondrial function (*isp-1*), reduced translation (*rsks-1, ife-2, ifg-1*), and germlineless animals (*glp-1*) all displayed smaller nucleoli in several tissues (Fig. 2c,d and Supplementary Fig. 2d,e). *ncl-1* mutation variously suppressed nucleolar size in these backgrounds (Supplementary Fig. 2b–d). The FOXO homolog *daf-16*, which promotes *daf-2* longevity, was also required for small nucleolar size of *daf-2* mutants, supporting the notion that these signalling pathways impinge on the nucleolus to regulate longevity (Fig. 2c,d and Supplementary Fig. 2e).

Isogenic wildtype worms show considerable variability in life expectancy, with some animals dying as early as day 10 and

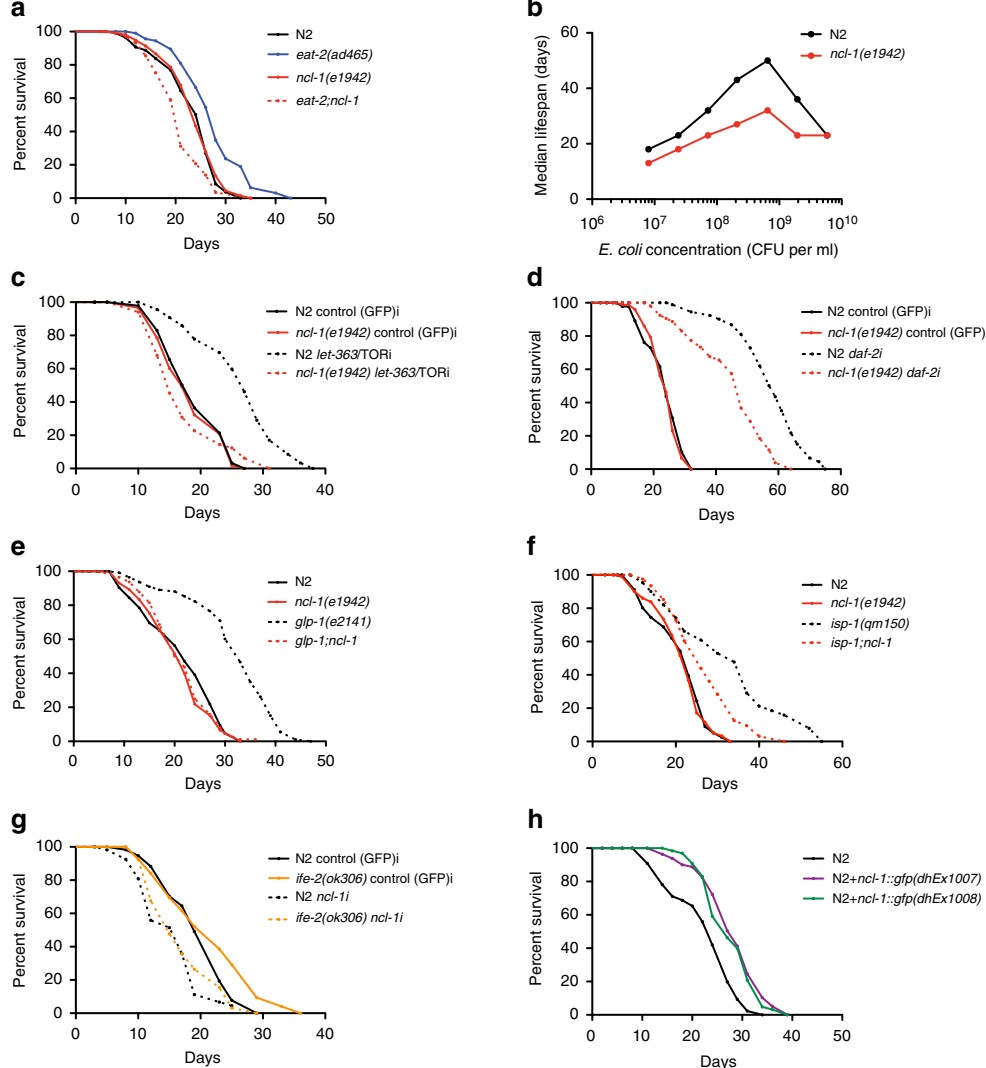

**Figure 1 | *ncl-1* mediates DR and other forms of longevity.** (**a**) Longevity of *eat-2(ad465)* is abolished with the loss of *ncl-1(e1942)* (*P* < 0.0001, three independent biological replicates). (**b**) *ncl-1(e1942)* is significantly shorter lived than N2 on bacterial dilution across 7 different concentrations (*P* < 0.0001, three independent biological replicates). (**c,d**) *ncl-1(e1942)* is shorter lived than N2 on *let-363*/TOR and *daf-2* RNAi (*P* < 0.0001, three independent biological replicates). (**e,f**) *glp-1(e2141)* and *isp-1(qm150)* are significantly longer lived than *glp-1;ncl-1* (*P* < 0.0001, three independent biological replicates) and *isp-1;ncl-1* (P = 0.0016, three independent biological replicates) respectively. (**g**) *ncl-1* RNAi significantly shortens the longevity of *ife-2(ok306)* (*P* < 0.0001, two independent biological replicates) (**h**) Over-expression of *ncl-1(+)* in N2 for two independent extra-chromosomal transgenic arrays (*dhEx1007, dhEx1008*) increases lifespan (*P* < 0.0001, three independent biological replicates). *P*-values calculated by log-rank test.

others as late as day 30, despite culture in a uniform environment. The basis of this variability however has remained elusive. We also found that wild-type animals showed variability in nucleolar size, and therefore wondered if these differences associate with lifespan in wild-type populations. To address this question, we imaged the nucleoli of age-matched worms on the first day of adulthood, recovered them on single plates and monitored their lifespan individually (Fig. 2e). We found a striking inverse correlation (Pearson correlation coefficient, 0.61–0.93) between nucleolar size and longevity, where animals with smaller nucleoli lived considerably longer than the ones with larger nucleoli (Fig. 2f). Thus nucleolar size could be a source of variability in longevity and may predict *C. elegans* life expectancy.

**Longevity mutants have reduced ribosome biogenesis.** To unravel molecular mechanisms, we examined how *ncl-1* and various longevity mutants affected nucleolar functions. Loss of

*ncl-1* has been previously shown to up-regulate the nucleolar protein FIB-1/fibrillarin[22,23], which serves as a methyltransferase for pre-rRNA processing and modification, and regulates histone modification[9,24]. In accord with this, we also observed increased levels of FIB-1::GFP as well as endogenous FIB-1 in *ncl-1* mutants (Fig. 3a,c and Supplementary Fig. 3a,b). Conversely *ncl-1* over-expression down-regulated FIB-1 (Supplementary Fig. 3b). We next asked if FIB-1 expression was affected in various longevity mutants. Indeed both FIB-1::GFP and endogenous FIB-1 were significantly reduced in *eat-2, daf-2, glp-1, isp-1* mutants and on TOR knockdown, and loss of *ncl-1* reversed this effect (Fig. 3a,c,d and Supplementary Fig. 3a,c), revealing that these pathways converge on FIB-1 expression. We further asked if FIB-1 is a passive marker or a causal factor for longevity. Consistent with the latter, *fib-1* RNAi knockdown reduced nucleolar size and extended lifespan of wild-type worms (Fig. 3e,f). RNAi knockdown of another gene involved in nucleolar function, *rrn-3*, which encodes TIF1A that assists in rRNA transcription

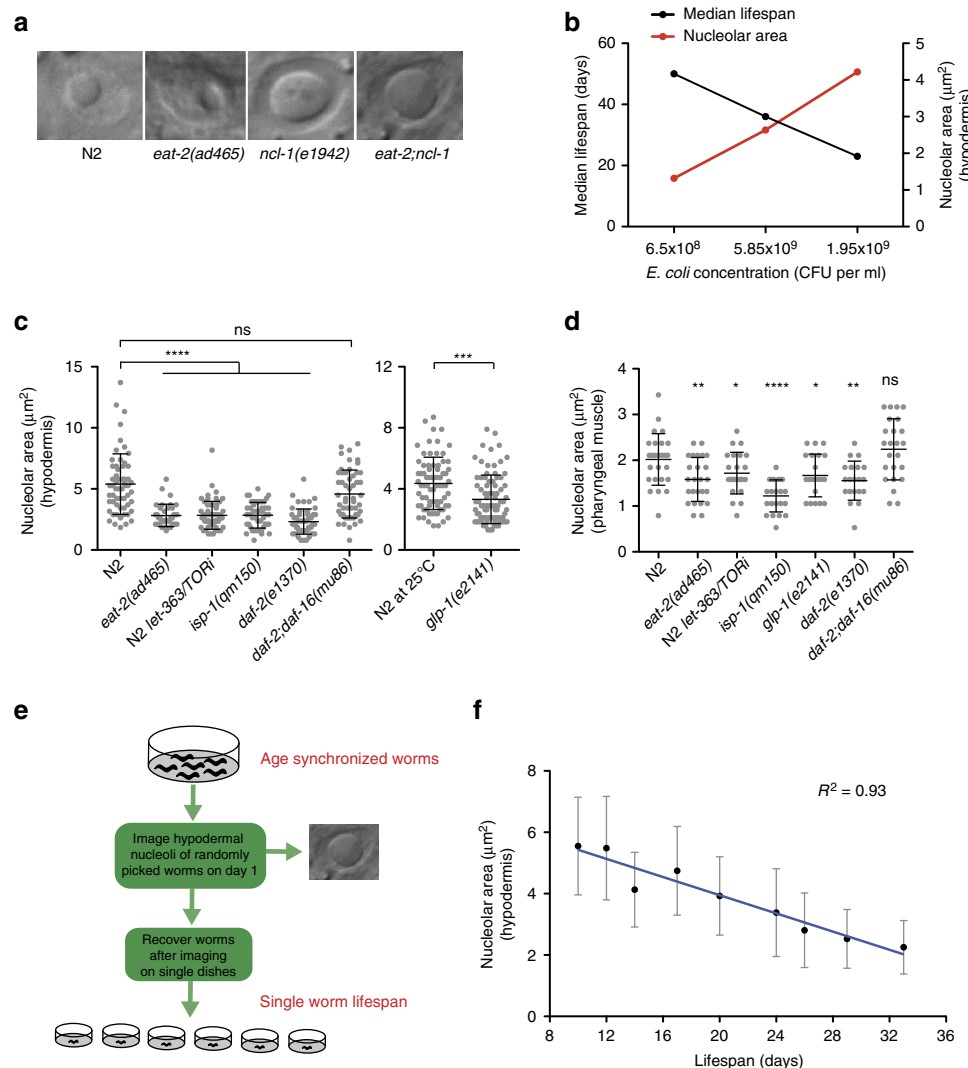

**Figure 2 | Nucleolar size inversely correlates with longevity.** (**a**) *eat-2(ad465)* animals have smaller nucleoli while *ncl-1(e1942)* and *eat-2;ncl-1* animals possess larger nucleoli compared to N2. (20 worms imaged per replicate, 3 independent biological replicates) (**b**) Nucleolar size is reduced on bacterial food reduction with a corresponding increase in lifespan. ($P < 0.0001$, log-rank test). (**c,d**) *eat-2(ad465)*, TOR RNAi, *isp-1(qm150)*, *glp-1(e2141)* and *daf-2(e1370)* animals possess smaller nucleoli while *daf-2;daf-16* have nucleoli similar to N2 in the hypodermis and pharyngeal muscle (Error bars represent mean ± s.d.) (**e,f**) Schematic illustrating the experiment, which shows that longer-lived worms exhibit small nucleoli and vice versa. (The graph depicts mean and s.d. Pearson correlation coefficient $R^2 = 0.93$ is calculated using the entire data set, the line equation is $y = -0.1483x + 6.9148$). Scale bar represents 5 μm. *$P < 0.05$, **$P < 0.01$, ***$P < 0.001$, ****$P < 0.0001$, NS, non-significant, unpaired *t*-test.

mediated by RNA Polymerase I[9], had little observable effect on longevity, perhaps because achieving a balance where benefits outweigh deleterious effects is difficult (Supplementary Fig. 3j).

Nucleoli are the cellular site of ribosome biogenesis. We therefore examined the expression levels of rRNA and ribosomal proteins. Mutation of *ncl-1* increased rRNA and ribosomal protein levels (Fig. 3b,c and Supplementary Fig. 3f–h). These molecules were also reduced in worms over-expressing *ncl-1* (Supplementary Fig. 3d,e). Notably long-lived *eat-2*, *daf-2*, *glp-1*, *isp-1* and TOR RNAi knockdown worms exhibited reduced levels of rRNA and ribosomal proteins RPS6 and RPS15, suggesting down-regulated ribosome biogenesis associates with longevity (Fig. 3b,c and Supplementary Fig. 3f–h). Loss of *ncl-1* variously suppressed these phenotypes in double mutant backgrounds (Fig. 3b,c and Supplementary Fig. 3f–h). Similarly, *daf-16* mutation restored the reduced rRNA and ribosomal proteins levels seen in *daf-2* mutants back to wild-type levels (Fig. 3b and Supplementary Fig. 3f,i). Taken together, these results suggest

that smaller nucleoli, reduced fibrillarin and ribosome biogenesis, are signatures of long life.

**Smaller nucleoli associate with longevity in higher organisms.** Given our results in *C. elegans*, we wondered if these findings hold true in long-lived models in other species. Remarkably we found that long-lived *Drosophila melanogaster* undergoing DR, exposed to the mTOR inhibitor rapamycin, or harbouring deletion of the insulin-like peptides *ilp-2-3,5*, all had smaller nucleoli in the fat body and intestine (Fig. 4a,b). Furthermore, they showed reduced levels of fibrillarin and ribosomal proteins, although RPS6 and RPS15 levels did not significantly change on Rapamycin treatment in flies unlike worms (Supplementary Fig. 4a–d). Age-matched mice undergoing DR and long-lived IRS1 knockout mice also exhibited smaller nucleoli in kidney, liver and whole brain sections compared to controls (Fig. 4c,d and Supplementary Fig. 4e–g). Finally, we also observed an overall trend towards reduction of nucleolar size in muscle biopsies of

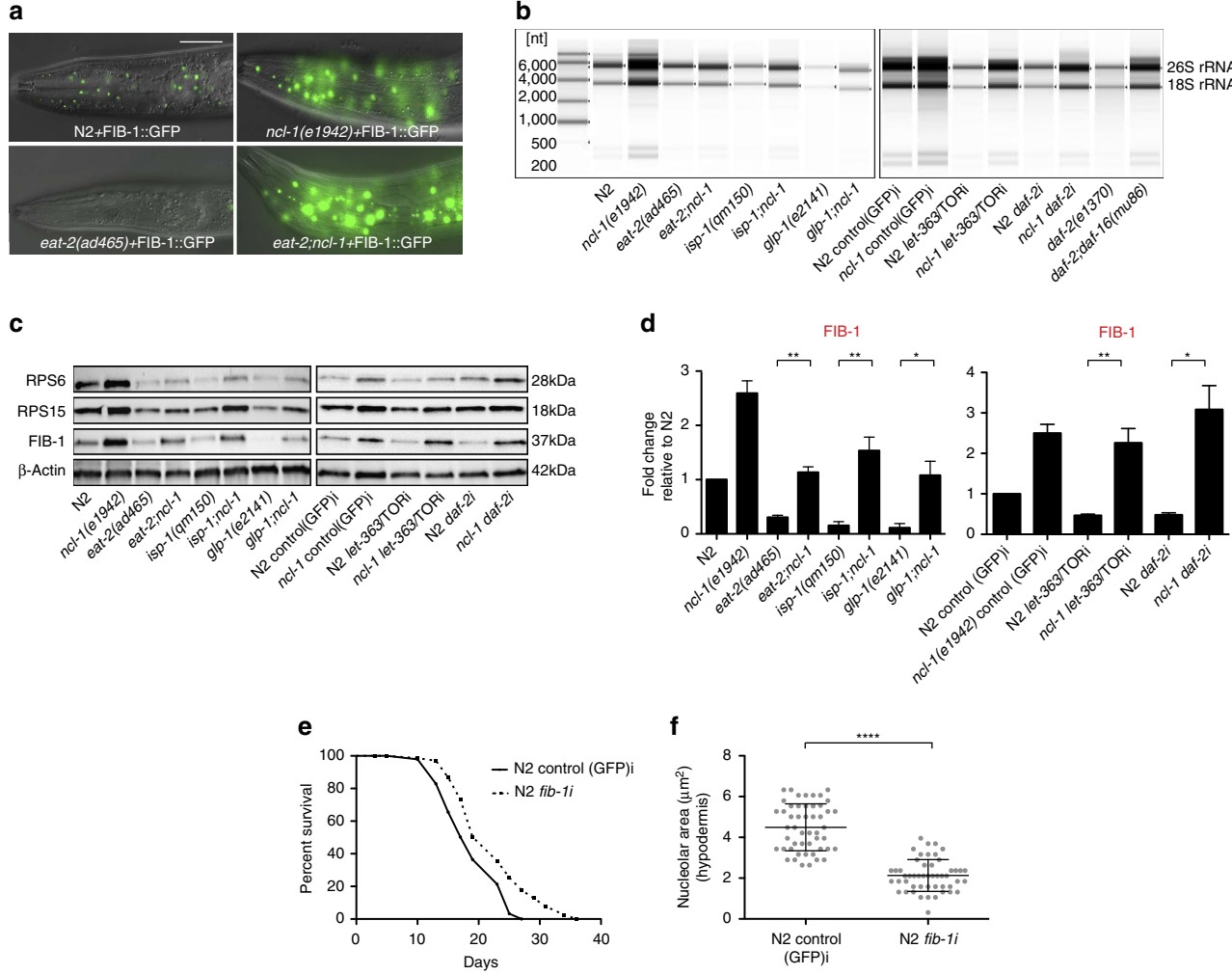

**Figure 3 | Longevity mutants have reduced ribosome biogenesis.** (**a**) FIB-1::GFP is strongly down-regulated in *eat-2(ad465)* animals but up-regulated in *ncl-1(e1942)* and *eat-2;ncl-1* double mutants (20 worms imaged per replicate, 3 independent biological replicates). (**b–d**) rRNA, RPS6, RPS15 and FIB-1 levels are increased in *ncl-1(e1942)* and reduced in *eat-2(ad465)*, *isp-1(qm150)*, *glp-1(e2141)*, *daf-2* RNAi and TOR RNAi and this effect is partially reversed by the loss of *ncl-1*. *daf-2(e1370)* also shows reduced rRNA levels compared to *daf-2;daf-16(mu86)* (Error bars represent mean ± s.e.m.). (**e**) *fib-1* RNAi extends lifespan of N2 (*P* = 0.0004, log-rank test, three independent biological replicates) (**f**) *fib-1* RNAi reduces the nucleolar size of N2 (Error bars represent mean ± s.d.). Scale bar represents 20 μm. *P<0.05, **P<0.01, ****P<0.0001, unpaired *t*-test.

elderly human volunteers who underwent a regime reducing caloric intake by 12.5% combined with moderate increase in exercise by 12.5% (Fig. 4e,f).

## Discussion

Altogether our studies reveal that multiple longevity pathways strikingly reduce nucleolar size, and diminish expression of the nucleolar protein FIB-1, ribosomal RNA, and ribosomal proteins across different species. A trend towards nucleolar size reduction is also seen with interventions that improve metabolic health in humans, thus revealing a reversible process linking metabolic state to a simple cellular readout. Conversely a parallel study reported that fibroblasts derived from Hutchinson-Gilford progeria syndrome patients show enlarged nucleolar size and elevated ribosome biogenesis and protein synthesis[25]. These markers are not simply molecular correlates, however, but likely responsible in part for prolonged life. Notably, *C. elegans* FIB-1 is regulated by multiple molecular pathways, and its down-regulation is sufficient to extend lifespan. Although knockdown

of the nucleolar RRN-3/TIF1A had little effect on lifespan, conceivably other nucleolar functions could play a role. Evidently NCL-1 is critical to regulating nucleolar size and inhibiting FIB-1 expression, thereby affecting lifespan in multiple pathways. How cytosolic NCL-1 impacts nucleolar function remains unclear, although evidence hints that it regulates FIB-1 in part via its 3′ UTR[23]. NCL-1 itself is not visibly regulated by longevity pathways (V. Tiku, personal communication); further studies should help unravel the mechanism of NCL-1 and FIB-1 action.

Our studies are among the first to reveal that nucleolar functions work pervasively across many longevity pathways. If small nucleoli are a hallmark for longevity, what are the proximal mechanisms responsible for extended life? Reduced ribosome biogenesis and protein synthesis are the most obvious candidates: These energetically costly processes consume considerable resources, and a modest reduction of ribosomal proteins or translational regulators in model organisms prolongs life[14,20,26,27]. Notably, transcriptomic analyses typically exclude ribosomal RNA, thus overlooking the very molecules that could well affect longevity. If protein synthesis alone were rate limiting, then knockdown of this

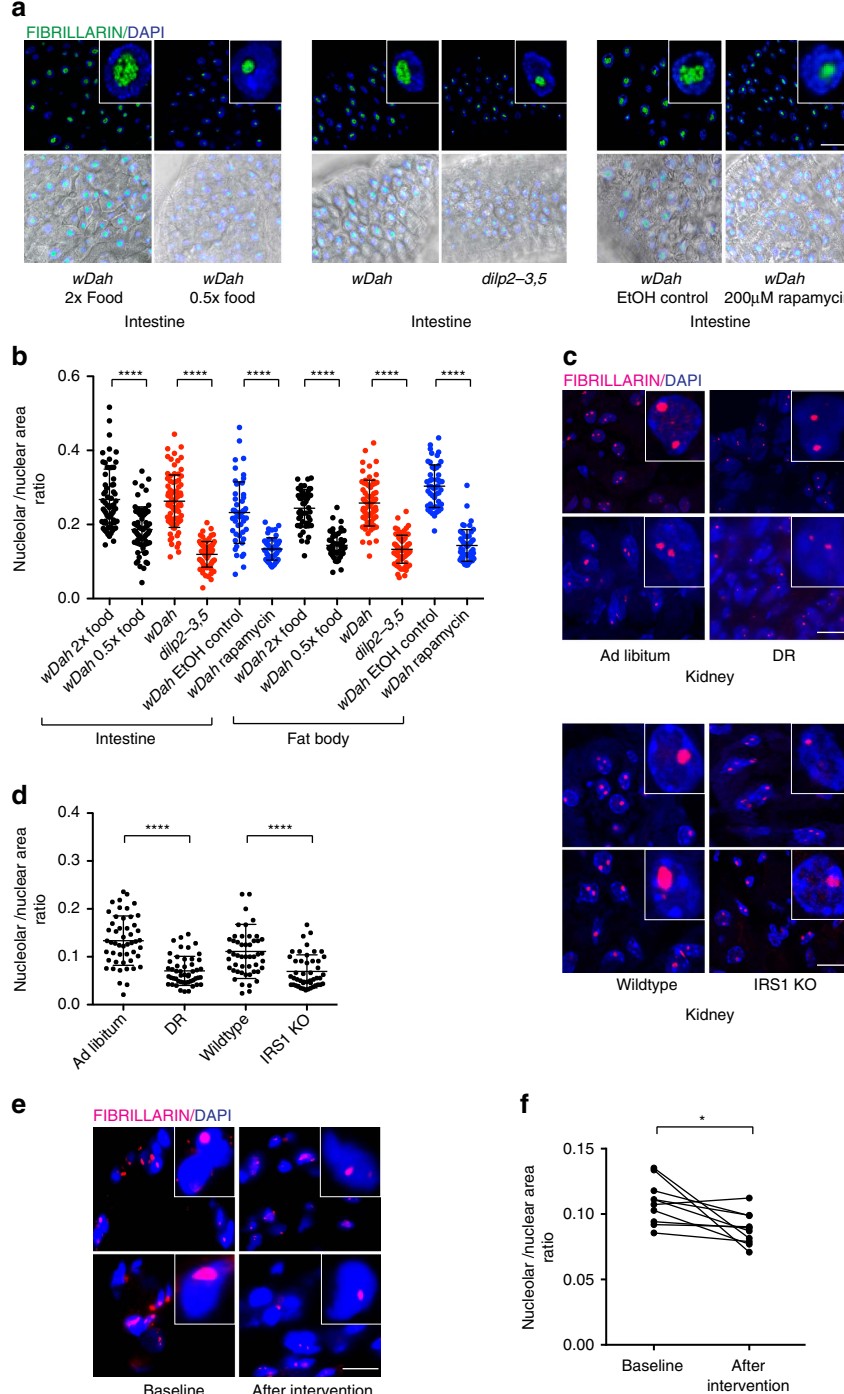

**Figure 4 | Smaller nucleoli associate with longevity in higher organisms.** (**a,b**) DR, *dilp2-3,5* and Rapamycin treated *D. melanogaster* possess small nucleoli in intestine and fat body (Error bars represent mean ± s.d.). (**c,d**) DR and IRS1 knockout mice show reduced nucleolar size in kidney tissue compared to ad libitum fed mice and wildtype (Error bars represent mean ± s.d.). (**e,f**) Muscle biopsies from humans undergoing DR and exercise exhibit small nucleoli. Scale bars represent 10 μm (**a,c**) and 20 μm (**e**). *$P < 0.05$ paired t-test, ****$P < 0.0001$ unpaired t-test.

process should restore longevity to *ncl-1* mutants. Surprisingly, it did not. Our genetic analysis reveals that mutants that reduce protein synthesis, namely *rsks-1*, *ife-2*, and *ifg-1*, prolong life in a *ncl-1* dependent manner, suggesting that NCL-1 works largely downstream or parallel to protein synthesis. This raises the interesting prospect that NCL-1 perturbs protein synthesis independently of these other factors, or that other cellular processes might be involved. Reduced protein synthesis *per se* may represent only one aspect of longevity, since lifespan extension by ribosomal inhibition triggers various aspects of the stress response[28,29]. In yeast recombination at rDNA repeats has been implicated in aging[30]. The nucleolus is also the site for assembly of other ribonucleoprotein particles including splicing complexes, telomerase, the signal recognition particle, stress granules and microRNA machinery, and regulates processes involved in genome integrity, nuclear architecture, stress signalling, cell cycle, and growth[9]. Conceivably these other nucleolar processes could also contribute.

Our work suggests that nucleolar size is a highly predictive marker for wild-type *C. elegans* longevity. Other markers that can approximate life expectancy and health status have been reported in *C. elegans*[31,32]. Our study opens up the exciting prospect that nucleolar size can predict life expectancy in higher organisms. If so, quantification of nucleolar size could be used as a single-cell readout of metabolic changes to study biological heterogeneity in aging and longevity, or to assess how various environmental and pharmacologic interventions impact health. We also imagine that there may well be exceptions or conditions, in which downstream processes uncouple nucleolar size from lifespan; *ncl-1* mutants themselves have enlarged nucleoli but near normal lifespan. In the future it will be fascinating to dissect the mechanisms underlying NCL-1 and FIB-1 action, the proximal nucleolar functions critical for longevity, and to further explore nucleolar functions as biomarkers of health and lifespan.

## Methods

***C. elegans* strains.** All the worm strains were grown using standard procedures at 20 °C unless otherwise noted[33]. Strains carrying *glp-1(e2141)* mutation were maintained at 15 °C and shifted to 25 °C for inducing germlineless phenotype. The strains used for the experiments were: N2 (wild type), *eat-2(ad465)*, *ncl-1(e1865)*, *ncl-1(e1942)*, *eat-2(ad465);ncl-1(e1865)*, *eat-2(ad465);ncl-1(e1942)*, *isp-1(qm150)*, *isp-1(qm150);ncl-1(e1942)*, *glp-1(e2141)*, *glp-1(e2141);ncl-1(e1942)*, *daf-2(e1370)*, *daf-2(e1370);daf-16(mu86)*, *cguIs001* (FIB-1::GFP)[34], *eat-2(ad465) + cguIs001*, *ncl-1(e1942) + cguIs001* and *eat-2(ad465);ncl-1(e1942) + cguIs001*, *ife-2(ok306)*, *ifg-1(cxTi9279)* and *rsks-1(sv31)*. dhEx1007 and dhEx1008 *ncl-1* extra-chromosomal transgenic strains were generated by injecting fosmid DNA WRM0611AC10 (*ncl-1*::TY1 EGFP) (30 ng μl$^{-1}$) and a co-injectable marker (*myo-2::mcherry* at 10 ng μl$^{-1}$) in N2 strain and further crossed into *eat-2(ad465)*, *eat-2(ad465);ncl-1(e1865)* and *eat-2(ad465);ncl-1(e1942)* backgrounds. The transgenic worms were maintained by selecting the worms showing the expression of the co-injected marker.

**Lifespan analyses.** All the lifespan analyses experiments were performed in at least three independent biological replicates at 20 °C. Animals that crawled off the plates, burst due to a ruptured vulva or had internal hatching of the eggs were censored from the experiment. RNAi lifespan analysis experiments were carried out following previously described protocol[35]. All RNAi treatments were performed throughout development and adulthood except *let-363*/TOR and *fib-1*, which were initiated on the first day of adulthood. For BDR lifespan analyses, the method followed was the same as described previously[36]. 90 worms were used for each bacterial concentration to be tested and the worms were scored every 3–4 days. The worms were transferred to freshly prepared bacterial conditions on each day of scoring. BDR medium containing FUdR (1 μg ml$^{-1}$) was used for the first two weeks of the experiment to prevent progeny production. All the lifespan experiments were performed in a blinded manner and repeated at least three times except for *ife-2(ok306)* (performed twice), *rsks-1(sv31)* (performed twice), *ifg-1(cxTi9279)* (performed once) and *rrn-3* RNAi (performed once). Mantel-Cox Log Rank method was used for statistical analysis (Supplementary Table 1).

**rRNA analysis.** Age-matched day 1 adult worms were collected in TRIzol (Invitrogen) and snap-frozen in liquid nitrogen. RNA extraction was performed using RNeasy Mini kit (QIAGEN). Levels of rRNA were analysed by running total RNA, extracted from the same number of worms on Agilent 4200 TapeStation System following High Sensitivity RNA ScreenTape System protocol (Agilent). rRNA levels were also examined by running total RNA extracted from the same number of worms on agarose gels. NorthernMax Kit protocol was followed for running RNA gels. The gels were imaged with Alpha Innotech MultiImage II.

(For RNA extraction: *n* = 100 worms/replicate, 3 independent replicates)

**Western blotting.** Day 1 adult worms (50) were collected in Laemmli lysis buffer and snap-frozen in liquid nitrogen. The samples were then boiled at 95 °C for 5 min, ultrasonicated for 10 cycles and loaded on 4–15% Mini-PROTEAN TGX Precast Protein Gels. After separation, proteins were blotted on a nitrocellulose membrane and probed with the following antibodies against: RPS-6 (Abcam ab70227, 1:1,000), RPS-15 (antibodies-online.com ABIN503870, 1:1,000), Fibrillarin (Novus Biologicals NB300-269, 1:1,000) and β-Actin (Abcam ab8224, 1:5,000). The uncropped western blotting images are shown in Supplementary Fig. 5.

(For all western blots: *n* = 50 worms/replicate, 3 independent replicates)

For *Drosophila* western blots, 5 females were homogenized in 100 μl of RIPA lysis buffer carrying 1X Complete mini protease inhibitor (EDTA free) (Roche). Extracts were cleared by centrifugation and protein content determined with BCA assay. 30 μg of total protein was loaded on precast gels (Bio-Rad Any KD, Mini-PROTEAN TGX). The proteins were transferred to nitrocellulose membranes and probed with the same antibodies as above. The uncropped western blotting images are shown in Supplementary Fig. 5. (For all western blots: *n* = 5 flies/replicate, 3 independent replicates).

**Immunofluorescence.** Immunofluorescence was performed on 10 μm thick cryo-sections of mouse tissues derived from kidney, liver and brain. The samples were fixed with 4% Paraformaldehyde (PFA) for 15 min at room temperature (RT) followed by three washes with PBS at RT. The samples were then blocked with 5% Normal Donkey Serum in PBS with 0.1% Triton-X for 30 min at RT followed by an over-night incubation at 4 °C with the primary antibody against Fibrillarin (Abcam ab166630, 1:200). After three subsequent washes with PBS, the samples were then probed with the secondary anti-rabbit antibody at RT for one hour followed by three more washes with PBS. The samples were mounted with ProLong Gold Mounting Medium (ThermoFisher Scientific). Immunofluorescence quantification represents three independent biological replicates with each replicate representing 3 mice (DR) and 2 mice (IRS1 KO). Imaging and quantification of the experiments were performed in a blinded manner.

*Drosophila* guts and fat bodies were dissected out in PBS followed by immediate fixation with 4% PFA in PBS and permeabilization for 10 min at RT with 0.3% Triton X-100 in PBS (PBST). Blocking, primary and secondary antibody incubation were done in 5% BSA in PBST using Fibrillarin (Novus Biologicals NB300-269, 1:250) as the primary antibody and goat anti-mouse conjugated to Alexa Fluor 488 (Invitrogen, Inc., 1:1,000) as the secondary antibody. Hoechst 33342 was applied at 1:1,000 for staining nuclei. Tissues were extensively washed with PBST after antibody treatments and finally mounted on glass slides with 80% glycerol in PBS. The quantification represents three independent biological replicates with each replicate representing 5 dissected flies. Imaging and quantification of the experiments were performed in a blinded manner.

For staining human muscle biopsies, samples were thawed at RT. Then the samples were blocked with 5% milk in PBS with 0.05% Tween (PBST) for 30 min at RT, followed by three washes with PBST. The primary antibody, Rabbit-anti-Fibrillarin (Abcam ab166630, 1:600 in PBST), was incubated overnight at 4 °C. After three washes with PBST, samples were incubated with the secondary goat-anti-rabbit-conjugated-Alexa647 antibody (Molecular Probes, 1:1,000 in PBST) for 1 h at RT, followed by three washes in PBST and one wash in PBS containing DAPI (0.5 μg ml$^{-1}$, Sigma-Aldrich, Saint Louis, Missouri, USA). Slides were mounted with Aqua Poly-Mount (Polysciences Inc, Niles, Illinois, USA). All samples were stained on the same day with the same antibody mixes.

**Imaging and quantification.** DIC microscopy was used to perform all the nucleolar imaging. Hypodermal, germ cell and pharyngeal muscle nucleoli of age-matched day 1 adults were imaged using 100X magnification with Axio Imager Z1 (Zeiss). Nucleolar area was quantified manually with the freehand tool using Fiji software. Details of the nucleolar size analysis are given in Supplementary Table 2. Worms carrying FIB-1::GFP and NCL-1::GFP transgenes were imaged using 63X magnification with Axio Imager Z1 (Zeiss). Immunofluorescent images were acquired using a laser-scanning confocal microscope (TCS SP5-X; Leica), equipped with a white light laser, a 405- diode UV laser, and a 100 × objective lens (HCX Plan-Apochromat CS 100 × oil, 1.46 NA). For human muscle biopsies, a total 15 representative fields with a 63X objective from each muscle sample were obtained, using the DM5500 fluorescent microscope (Leica) and the LAS AF software (version 2.3.6, Leica). Anti-Fibrillarin was detected with the Y5 cube, and nuclei were detected with the A4 cube. The area of the nucleolar and nuclear regions was quantified manually with the freehand tool, and subsequently the ratio of nucleolar/nuclear area was calculated. For the human samples, the average ratio of nucleolar/nuclear area (from an average of 100.4 ( ± 28.9) nuclei) per sample was used for the analyses.

***Drosophila melanogaster* experiments.** DR in *Drosophila melanogaster* was performed by feeding a total of 50 hatched flies with 0.5x SYA food compared to ad libitum food supply of 2 × SYA for 10 days[37]. Rapamycin treatment was performed by dissolving Rapamycin in absolute ethanol and mixing it with SYA food at a final concentration of 200 μM and fed to a total of 50 age-matched flies. For control food, ethanol alone was added. Both DR and Rapamycin treatment were performed for 10 days before harvesting the flies for experiments. The treatments were performed separately in 3 different vials serving as 3 independent biological replicates. Long-lived *dilp2-3,5* (ref. 38) and control *wDah* flies were harvested on day 1 of adulthood. The flies were dissected and immunofluorescence was performed on the dissected tissues as described above.

**DR and IRS1 KO mice.** Mouse experiments were performed according to the guidelines and approval of LANUV [Landesamt für Natur, Umwelt und Verbraucherschutz Nordrhein-Westfalen (State Agency for Nature, Environment and Consumer Protection North Rhine-Westphalia), VSG 84-02.04.2013.A158]. C57BL/6 male mice, obtained from Charles River Laboratories (Sulzfeld, Germany) were maintained under 12 h light:12 h dark schedule and were fed standard chow diet (SC)—4.5 g SC/animal/24 h (sniff Spezialdiäten GmbH) until 10 weeks of age and then subjected to DR at 75% food intake (3 g SC per animal/24 h) compared to ad libitum fed control mice. The DR regimen was continued for 1 month and the

mice were killed at the age of 14 weeks along with the ad libitum fed controls to perform cryo-sectioning for the analysis of nucleoli. The tissues sampled with sectioning were kidney and liver.

C57BL/6 IRS1 KO male[39] and WT control male mice were maintained similarly on SC diet. The animals were killed at the age of 12 months to perform cryo-sectioning for the analysis of nucleoli. The tissues sampled with sectioning were kidney and brain. C57BL/6 IRS1 KO mice were originally obtained from Prof. Dominic Withers' lab (Imperial College, London) and were bred on-site at the mouse facility in Max Planck Institute for Biology of Ageing, Cologne.

For both the experiments cryo-sectioning was performed horizontally across the entire tissue. This nature of processing aided in observing the effect of the treatments across different cell types in each tissue.

**Dietary restriction and exercise intervention in human volunteers.** Samples for nucleolar staining were obtained from the biomaterial collected in the Growing Old Together Study, a 13-weeks lifestyle intervention in older adults, consisting of 12.5% caloric restriction and 12.5% increase in physical activity, resulting in an average weight loss of 3.3 kg. The study design, inclusion and exclusion criteria, and changes in metabolic parameters have been described previously[40]. For the current study we used samples from 5 men and 5 women selected based on the greatest weight loss due to the intervention and the availability of muscle tissue from before and after the lifestyle intervention. This subgroup had an average age of 62.4 years ( ± 4.1) and lost an average of 6.8 kg ( ± 1.3) due to the intervention. Characteristics of this subgroup are detailed in Supplementary Table 3.

All participants signed a written informed consent for participating in this study. All experiments were performed in accordance with the relevant regulations and guidelines. The medical ethical committee of the Leiden University Medical Center approved this study. This trial (NTR3499) was registered at the Dutch Trial Register (www.trialregister.nl).

**Muscle biopsies and sectioning.** Muscle biopsies were collected from the *vastus lateralis* muscle before and after the lifestyle intervention. Biopsies were collected 40–45 min following a standardized liquid meal (Nutridrink, Nutricia Advanced Medical Nutricion, Zoetermeer, The Netherlands) in the morning after at least 10 h of fasting. Under local anaesthesia, an incision was made 10 cm cranial of the patella on the lateral side of the upper leg. A biopsy needle (3 mm thick) was inserted to obtain the muscle biopsy. The muscle biopsy was immediately frozen in liquid nitrogen and stored at − 80 °C before cryosectioning. Cryosections of 16μm were made with the CM3050-S cryostat (Leica, Wetzlar, Germany), pasted on SuperFrost Plus slides (Menzel-Gläser, Braunschweig, Germany) and stored at − 20 °C before staining.

**Blinding of experiments.** All the lifespan analysis experiments were performed in a blinded manner. For blinding, the strain names were concealed during scoring, analysing and plotting the data. Nucleolar imaging and quantification were also performed with concealed strain names.

*Drosophila* nucleolar size analysis was performed in a blinded manner. Two different people were involved in performing the experiment. One individual carried out fly feeding and mutant strain maintenance and the samples were passed on blinded for imaging and quantification to the second experimenter.

Mouse nucleolar size analysis was also carried out blinded. Three different people were involved in performing the experiments. One experimenter maintained the mice while carrying out the feeding/treatments and killed the mice for sectioning. The experiment was blinded henceforth. The sectioning was carried out blinded by the second experimenter. The blinded sections were stained, imaged and quantified by the third experimenter.

Two different experimenters performed human muscle biopsy staining. The whole experiment including staining, imaging and image quantification were performed completely blinded.

**Data availability.** The authors declare that all the data and the methods used in this study are available within this article, its Supplementary Information files, the peer-review file, or are available from the corresponding author on request.

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

## Acknowledgements

We thank S.J. Lo (Chang Gung University) for the FIB-1::GFP strain and the *Caenorhabditis* Genetics Center for other strains. We also thank Joana Goncalves and Kathrin Riehl for their assistance with mouse tissue sectioning, Dr Yidong Shen (Chinese Academy of Science) for his help in generating *ncl-1* transgenic lines, Dr Julia Noack and Dr Marian Beekman for help with data analysis. This work was supported by the Max Planck Society, CECAD/Deutsche Forschungsgemeinschaft (DFG), and Cologne Graduate School of Ageing Research doctoral scholarship (V.T.). The Growing Old Together Study received funding from the European Union's Seventh Framework Program (FP7/2007–2011), grant agreement number 259679.

## Author contributions

V.T., S.N. and A.A. designed experiments. V.T., C.J., B.H., W.L. and M.S. performed the experiments. Y.R. and H.E.D.S. performed stainings on human muscle biopsies. R.-U.M., P.E.S. and L.P. provided useful ideas in designing experiments. V.T. and Y.R. performed data analysis. V.T. and A.A. wrote the manuscript.

## Additional information

**Competing interests:** The authors declare no competing financial interests.

**Publisher's note**: 

