## [Peer Review File · Nature Communications]

Reviewers' Comments:

Reviewer #1 (Remarks to the Author)

In this study, Tiku et al. describe a relationship between nucleolar size lifespan in *C. elegans* and to a more limited extent in other species. Smaller nucleoli corresponds to longer lifespan generally and in worm is required or partially required for lifespan extension by a number of classic longevity pathways. These findings lead the authors to speculate that control of nucleolar size is a critical process for modulation of lifespan.

The findings are interesting but it is debatable how much they further the understanding of links between control of rDNA synthesis and lifespan, particularly for a high impact journal. Nucleolar size may be a reflection of reduced rRNA production and in turn reduced ribosome biogenesis. Reduced protein synthesis has been linked to lifespan on multiple occasions.

The authors also mention that recombination at rDNA repeats affects aging in yeast, but don't discuss potential mechanistic links including noncoding RNAs and competition between replication origins in rDNA versus the rest of the genome. While the authors are likely onto a finding of major significance, further studies need to be performed to delineate potential mechanisms why nucleolar size regulates lifespan.

On a minor issue, it is not clear to what extent *ncl-1* mutants block lifespan extension by calorie restriction as mediated by food dilution. Lifespan is shorter in the *ncl-1* mutant, but still affected by dilution. The block seems partial at best.

Reviewer #2 (Remarks to the Author)

The authors have addressed nucleolar size in the context of lifespan, mostly in worms, with supporting evidences in fruit flies, mouse, and humans. The main conclusion of the work is that there is a striking correlation between the size of the nucleolus and life expectancy: the smaller the nucleoli, the longer the animal live. This is quite an interesting finding. However, globally, the work remains very descriptive at this stage.

The authors show that a mutation in *ncl1*, previously linked to longevity in worms, leads to the overexpression of Fibrillarin, a box C/D snoRNA-associated methyltransferase involved in pre-rRNA processing and pre-rRNA modification (2'-O methylation), and increased accumulation of mature rRNAs. The authors show that depleting Fibrillarin in worms expand lifespan. It is not clear at this stage whether the effects reported with Fibrillarin on lifespan are specific, or whether they would be seen with any other ribosomal assembly factors (or translation factors). This should be addressed.

To get initial mechanistic insights, it would be very interesting to test if the variation in Fibrillarin' expression impacts rRNA methylation. In particular, there are hypomodified sites of 2'-O methylation that could be affected, leading to the production of specialized ribosomes with potentially differential functions in translation. Translational reprogramming could contribute to some of the effects observed on extended lifespan.

Specific comments:

-Fig 3, panels b, and c: how was inherent variability addressed here. The experiment shown is one observation of how many repeats? The bioanalyzer electropherograms shown in panel b are a good indication, they should be strengthened by a detailed pre-rRNA processing analysis by quantitative Northern blotting. There are bands additional to 28S and 18S rRNA on the profiles, what are these?

Panel e, isn't it a bit counterintuitive, considering that Fibrillarin is required for early pre-rRNA processing steps leading to the synthesis of 18S rRNA and to small ribosomal subunit production that knocking it down increases lifespan? The lifespan expansion observed is interesting but how specific is it of Fibrillarin? Would this be seen with other ribosome biogenesis factors?

-Fig 4 e, f: the experiment on human samples is very appreciated, but is the sampling enough to be really conclusive, should the authors rather discuss their observations in terms of a 'trend'.

-in their Discussion: when the authors discuss that the nucleolus is also required for the assembly

of other ribonucleoprotein complexes, most relevant to their work on aging is the notion that Telomerase is assembled in the nucleolus. This would be a useful addition in a future version of this manuscript.

-Fig 2f, showing an inverse correlation between nucleolar size and lifespan in a natural population of worms is good. It would be better to use the term 'variability' rather than 'variance' in the mainstream text (variance is used in statistics to express variability).

Reviewer #4 (Remarks to the Author)

In this work, Tiku, et al. show that nucleolar size inversely correlates with lifespan in *C. elegans*, that nucleolar size is a good biomarker and predictor of *C. elegans* lifespan, and that other organisms (flies and human cells) also show a correlation between longevity-regulating conditions (dietary restriction) and decreased nucleolar size.

In general, this short manuscript is clear, the experiments are well-executed, and the conclusions are striking. My major question is whether there are particular rRNA transcripts that are affected by nucleolar size that would then account for longevity differences. I would like to see an experiment that identifies the rRNA transcripts that are changed in the *ncl-1::gfp* transgene lines and a comparison with those in the *eat-2* vs *eat-2;ncl-1* lines to see if there are some transcripts that are the most affected by nucleolar size.

Reviewer 1.

Comment: The findings are interesting but it is debatable how much they further the understanding of links between control of rDNA synthesis and lifespan, particularly for a high impact journal. Nucleolar size may be a reflection of reduced rRNA production and in turn reduced ribosome biogenesis. Reduced protein synthesis has been linked to lifespan on multiple occasions. The authors also mention that recombination at rDNA repeats affects aging in yeast, but don't discuss potential mechanistic links including noncoding RNAs and competition between replication origins in rDNA versus the rest of the genome. While the authors are likely onto a finding of major significance, further studies need to be performed to delineate potential mechanisms why nucleolar size regulates lifespan.

Response: *While we agree with the reviewer that a clear molecular mechanism is lacking behind lifespan regulation and nucleolar size/function, we want to mention that the striking finding of a correlation between reduced nucleolar size and longevity across a host of different organisms is profound and fascinating. Even though it is known that a reduction in translation extends lifespan, it cannot be ruled out that other nucleolar functions are not involved. As part of our revision we investigated genetic epistasis between ncl-1 and translation downregulation in the context of lifespan regulation and nucleolar size. We observed smaller nucleoli in long-lived worms with reduced translation (ife-2,ifg-1,rsks-1) (see below), adding more evidence to our correlation of smaller nucleolar size and extended lifespan (Supplementary Fig 2d). Furthermore we also found that ncl-1 is required for longevity conferred by down-regulating translation (ife-2,ifg-1,rsks-1) (Fig. 1g and Supplementary Fig. 1e,f) (see below). Therefore ncl-1 is epistatic over translation suppression mediated longevity. In our future studies we are systematically investigating different nucleolar functions in order to establish a molecular connection between the nucleolus and lifespan regulation. More specifically, we will identify interaction partners of NCL-1 in order to get a better understanding of the downstream signaling. We will also employ unbiased transcriptomics and proteomics approaches to examine transcriptional and proteomic changes in ncl-1 mutants. These approaches will unravel the mechanism of how nucleolar function plays into lifespan regulation. However, this is a whole new study and beyond the scope of the current project.*

Comment: On a minor issue, it is not clear to what extent *ncl-1* mutants block lifespan extension by calorie restriction as mediated by food dilution. Lifespan is shorter in the *ncl-1* mutant, but still affected by dilution. The block seems partial at best.

Response: In our food dilution induced dietary restriction (DR) data *ncl-1* mutants (both null alleles) are not short lived (Fig 1b and Supp. Fig. 1b). In fact, *ncl-1* mutants have a lifespan comparable to wildtype N2 worms (Fig. 1 and Supp. Fig 1). Our DR lifespan data using both *eat-2* mutants and food dilution clearly show a complete abrogation of longevity (more so in *eat-2* where *eat-2;ncl-1* shows even a synthetic phenotype with both *ncl-1* null alleles [Fig. 1a and Supp. Fig. 1a]).

Reviewer 2.

Comment: The authors show that a mutation in *ncl1*, previously linked to longevity in worms, leads to the overexpression of Fibrillarin, a box C/D snoRNA-associated methyltransferase involved in pre-rRNA processing and pre-rRNA modification (2'-O methylation), and increased accumulation of mature rRNAs. The authors show that depleting Fibrillarin in worms expands lifespan. It is not clear at this stage whether the effects reported with Fibrillarin on lifespan are specific, or whether they would be seen with any other ribosomal assembly factors (or translation factors). This should be addressed.

Response: We are in complete agreement with the reviewer. As asked, we performed lifespan analysis experiments by knocking down another nucleolar gene, *rrn-3* which encodes the worm homolog of TIF-1A. TIF-1A assists RNA Polymerase I in rRNA transcription. We did not observe changes in lifespan upon the knock down of *rrn-3* in wildtype, *ncl-1*, *glp-1* and *glp-1;ncl-1* mutant worms (Supplementary Fig. 3j) (see below). We also studied genetic epistasis in long-lived mutants with reduced translation (*ife-2*, *ifg-1*). Both *ife-2(ok306)* and *ifg-1(cxTi9279)* lose their extended lifespan with the loss of *ncl-1* (Fig. 1g and Supplementary Fig. 1e) (see above). Furthermore, we also tested the long-lived S6 Kinase mutant *rsk-1(sv31)*; *ncl-1* RNAi also shortens the lifespan of *rsk-1(sv31)* animals (Supplementary Fig. 1f) (see above). We also observed smaller nucleoli in long-lived worms with reduced translation (*ifg-1,ife-2*) and *rsk-1* mutants and this effect is reversed with the loss of *ncl-1* suggesting that *ncl-1* is epistatic over the effects of translation reduction on both lifespan and nucleolar size (Supplementary Fig. 2d) (see above). Furthermore these data add more evidence to our correlation of smaller nucleolar size and extended lifespan. Taken together these results suggest that *ncl-1* is indispensable for longevity in these long-lived mutants and indeed the signals from reduced translation also converge on *ncl-1* to bring about extended lifespan. Of note, *ncl-1* mutation has not previously been linked to longevity. From our previous screen to identify novel factors in lifespan regulation (Heestand et. al., 2013), we identified *ncl-1* as a potential player in DR mediated longevity but these data were not published. We characterized here the role of *ncl-1* as a novel mediator of longevity across different lifespan regulating pathways.

Comment: To get initial mechanistic insights, it would be very interesting to test if the variation in Fibrillarin' expression impacts rRNA methylation. In particular, there are hypomodified sites of 2'-O methylation that could be affected, leading to the production of specialized ribosomes with potentially differential functions in translation. Translational reprogramming could contribute to some of the effects observed on extended lifespan.

[REDACTED]

Comment: Fig 3, panels b, and c: how was inherent variability addressed here. The experiment shown is one observation of how many repeats? The bioanalyzer electropherograms shown in panel b are a good indication, they should be strengthened by a detailed pre-rRNA processing analysis by quantitative Northern blotting. There are bands additional to 28S and 18S rRNA on the profiles, what are these?

Response: *The experiments shown in Figure 3b and c have been repeated three times independently. The data shown in Figure 3b and c has been quantified as bar graphs (Fig. 3d, and Supp. Fig. 3f,g,h). Bar graphs show mean±s.e.m. relative to wildtype N2 control from three independent experiments. All the repeats show similar results. We agree with the reviewer that a detailed pre-rRNA processing analysis would strengthen our findings. Since rRNA processing is a very complex phenomenon with multiple processing steps and numerous transient, unstable intermediates including 32S, 30S, 21S, 12S etc, it would be very difficult to reproducibly catch minor changes that might be occurring between long and short-lived mutants across different processing events. To detect such events will require scaling up considerably, which is technically challenging as we are hand-picking age matched worms. Instead of northern blotting, we are planning to take RNA sequencing approach to investigate rRNA processing changes, which will take considerable effort and cannot be managed in the time frame for revision. The additional bands on the image in Figure 3b are tRNA/snRNA, which run below 18S rRNA. We note here that in all our rRNA bioanalyzer images across replicates we obtained sharp bands for 26S and 18S rRNA and 26S rRNA bands were almost twice as intense as 18S rRNA which indicates good quality intact RNA without any degradation. We also note here that we have changed 28S to 26S rRNA in the manuscript since *C. elegans* possess 26S instead of 28S rRNA. It was a typing error on our part.*

Comment: Panel e, isn't it a bit counterintuitive, considering that Fibrillarin is required for early pre-rRNA processing steps leading to the synthesis of 18S rRNA and to small ribosomal subunit production that knocking it down increases lifespan? The lifespan expansion observed is interesting but how specific is it of Fibrillarin? Would this be seen with other ribosome biogenesis factors?

Response: *A modest knockdown of protein synthesis increases lifespan in several species. Indeed this and the finding that *fib-1* knockdown behaves similarly is counterintuitive, but it suggests that slowing this process down could broadly enhance other quality control or stress resistance mechanisms. The effect of Fibrillarin might be specific because inhibition of *rrn-3* (TIF-1A) homolog doesn't affect lifespan even though it is an important player in ribosome biogenesis as discussed above.*

Comment: The experiment on human samples is very appreciated, but is the sampling enough to be really conclusive, should the authors rather discuss their observations in terms of a 'trend'

Response: *We agree with the reviewer and have discussed the observation as a trend with the human samples. We note here that we could not perform an extensive analysis with the human samples because of the limiting material that we had at hand.*

Comment: In their Discussion: when the authors discuss that the nucleolus is also required for the assembly of other ribonucleoprotein complexes, most relevant to their work on aging is the notion that Telomerase is assembled in the nucleolus. This would be a useful addition in a future version of this manuscript.

Response: *The point about telomerase has been added to the manuscript*

Comment: -Fig 2f, showing an inverse correlation between nucleolar size and lifespan in a natural population of worms is good. It would be better to use the term 'variability' rather than 'variance' in the mainstream text (variance is used in statistics to express variability).

Response: *Variability has been included instead of variance in the mainstream text discussing the correlation between lifespan and nucleolar size.*

Reviewer 4.

Comment: In general, this short manuscript is clear, the experiments are well-executed, and the conclusions are striking. My major question is whether there are particular rRNA transcripts that are affected by nucleolar size that would then account for longevity differences. I would like to see an experiment that identifies the rRNA transcripts that are changed in the *ncl-1::gfp* transgene lines and a comparison with those in the *eat-2* vs *eat-2;ncl-1* lines to see if there are some transcripts that are the most affected by nucleolar size.

Response: *We agree with the reviewer that a detailed landscape of rRNA transcripts of short-lived versus long-lived mutants would strengthen our findings. We, in theory could perform northern blot analysis, to explore such changes. However we have consulted several experts in the field, and the consensus is that northern blot analysis is not a very accurate and reliable method to examine rRNA processing intermediates. Since rRNA processing is a very complex phenomenon with multiple processing steps and numerous transient unstable intermediates including 32S, 30S, 21S, 12S etc, it would be technically challenging to reproducibly catch minor changes that might be occurring between long and short-lived mutants across different processing events. Instead of northern blotting, we are planning to take RNA sequencing approach to investigate rRNA processing changes, which lies outside of the scope of the paper, and would raise only further questions for analysis.*

Reviewers' Comments:

Reviewer #2:

Remarks to the Author:

The authors have adequately addressed my comments.

A full pre-rRNA processing analysis (also requested by another referee) and a full analysis of rRNA 2'-O-methylation status would have been ideal.

However, I understand that these are quite challenging at this stage. The manuscript has sufficient novelties in it to make it a worthwhile contribution.